# Insight into Body Condition Variability in Small Mammals

**DOI:** 10.3390/ani14111686

**Published:** 2024-06-05

**Authors:** Linas Balčiauskas, Laima Balčiauskienė

**Affiliations:** Nature Research Centre, Akademijos 2, 08412 Vilnius, Lithuania; laima.balciauskiene@gamtc.lt

**Keywords:** body condition index, reproduction, gender, age, small mammals

## Abstract

**Simple Summary:**

Based on a sample of nearly 30,000 individuals representing 18 small mammal species trapped in Lithuania between 1980 and 2023, we conducted the first multi-species analysis of the Body Condition Index (BCI) of small mammals in middle latitudes. Our analysis compared BCIs between species and examined differences in age groups, gender, and reproductive statuses within each species (seven species had sample sizes with N < 100 individuals). Among the species with the highest average BCI, seven out of eight granivores or omnivores consume animal food occasionally. Two distinct patterns in the BCI were identified during ontogeny, either decreasing or increasing from juvenile to adult, possibly related to differences in their diet. Our results demonstrate that reproductive stress has a negative impact on the BCI of adult females in all analyzed species and nearly all adult males. We observed a rare case of the Chitty effect in two species of shrews, where their high body mass resulted in a very high BCI. Our results help to understand how changing environmental conditions are affecting small mammals. This is the first multi-species approach of body condition at middle latitudes.

**Abstract:**

The body condition index (BCI) is an indicator of both reproductive success and health in small mammals and might help to understand ecological roles of species. We analyzed BCI data from 28,567 individuals trapped in Lithuania between 1980 and 2023. We compared BCIs between species and examined differences in age groups, gender, and reproductive statuses within each species. Seven out of eighteen species had sample sizes with N < 100. In terms of species, we found that seven of the eight species with the highest average BCIs are granivores or omnivores, which can consume animal-based food at least seasonally. The two contrasting (decreasing or increasing) BCI patterns observed during ontogeny can be related to diet differences among juveniles, subadults, and adult animals. Our results demonstrate that reproductive stress has a negative impact on the BCI of adult females in all analyzed species and nearly all adult males. Although the animals with extremely low BCI consisted mostly of shrews, for the first time we found 23 common and pygmy shrews exhibiting the Chitty effect, i.e., a very high body mass resulting in a BCI > 5.0. This is the first multi-species approach of body condition at middle latitudes. The results increase our understanding of how changing environmental conditions are affecting small mammals.

## 1. Introduction

The body condition of animals, often assessed through body condition indices (BCI), is considered a proxy for individual fitness [1,2]. Both traits are very important in ecological and evolutionary studies [2]. However, measuring fitness directly can be complex. Therefore, many studies rely on body condition, which is assessed using various indices. These indices are employed to investigate diverse topics such as life histories, reproduction, and conservation resource management [3,4].

In general, it is assumed that individuals should have a higher BCI to be successful, or, in other words, be heavier for the same body size, in which case even body mass alone is a good measure of an individual’s fitness [5]. Therefore, the concept of body condition is important as an ecological indicator, as the best conditioned animals survive better in harsh winter environments [6] and are better selected for breeding [7]. Winter is the most threatening time for small herbivores such as voles, because at a time of limited food resources they experience greatest energy requirements [6]. Starvation during this period reduces their survival; on the other hand, fat reserves increase not only survival but also subsequent reproductive success [8].

Shrews survive the winter by shrinking their body size and reducing body mass, i.e., decreasing their BCI–the so called Dehnel phenomenon [9,10]. Various species of voles increase their chances of survival in winter by maintaining optimal, not too big, body mass in late autumn to lessen food requirements during winter [6,11,12,13,14]. At mid-latitudes, we observed a winter growth depression in the common vole (*Microtus arvalis*), which occurred in January–February in juveniles and in January–March in subadults. The growth depression was more pronounced and of longer duration in terms of cranial traits than in terms of body mass [15].

In the postnatal development of small mammals, body condition likely plays a role in their reproductive success. However, in prairie voles (*Microtus ochrogaster*), body condition was found to have no influence on their reproduction or survival [16]. Therefore, it is agreed that mammalian life histories have evolved with consideration for their varying body sizes [17]. Furthermore, long-term changes in body mass, not just phenological responses, can indicate adaptation to climate change in middle and higher latitudes [18].

The quantification and analysis of body condition indices have been the subject of debate for decades [2]. While numerous indices have been tested and reviewed, relatively few have been applied to small mammals [3,8,19,20,21,22]. To ensure comparability, biases in body size and mass measurements among different investigators must be minimized, particularly in live-trapping studies [20].

Several studies have analyzed various factors important to the BCI of small mammals in different latitudes. These factors include food availability or the overall amount of food resources [23]. Food resources are closely linked to habitat and habitat changes induced by human activities. However, in the case of Tullberg’s soft-furred mouse (*Praomys tullbergi*), habitat influence was not pronounced [24]. Other studies in low latitudes have shown that habitat and habitat fragmentation affect the body condition of individuals [25,26]. Our current study will definitively fill the gap in BCI information from the middle latitudes.

Some researchers caution against misinterpreting habitat use as habitat preference [27], as the latter involves an individual’s preference for particular resources and conditions. We believe that, at the very least, time scales should be analyzed in conjunction with habitat use in order to make implications to fitness.

Body condition can be negatively affected by pollution, whether it is heavy metals in contaminated sites [28,29] or biogenic pollution in seabird colonies [30]. In the latter example, the BCI varied among zones within the great cormorant (*Phalacrocorax carbo*) colony, with the most active nesting areas showing notable differences.

Two important conclusions on the extremes of body condition have been drawn from studies on California vole (*Microtus californicus*) and meadow vole (*Microtus pennsylvanicus*). The first study indicates that the diet and body condition of dispersers, predominantly young males with body mass below the average, do not differ from resident individuals [31]. To our knowledge, the relationship between dispersal and BCI has not been investigated in European voles, except for two related studies on the bank vole (*Clethrionomys glareolus*) [32,33]. The second conclusion explains the presence of large animals in the population exhibiting the Chitty effect [34], which is crucial for understanding extreme BCI values.

The improved body condition of the yellow-necked mouse (*Apodemus flavicollis*) was attributed to supplementary feeding [35]. Larger individuals generally have better body condition and are less affected by parasite infections [36]. In the case of the striped field mouse (*Apodemus agrarius*), body size and body condition have been used as indicators of a disturbed environment [37]. In the wood mouse (*Apodemus sylvaticus*), BCI has been analyzed in relation to habitat quality [38], particularly focusing on the agro-silvo-pastoral Mediterranean landscape in Portugal [39].

It has been shown that *C. glareolus* females are more likely to reproduce and improve their body condition when supplemented with high quality food, resulting in larger litters [40]. This study confirmed the results of A.F. Karlsson [41] that body weight alone cannot predict overwintering success in this species.

In root voles (*Alexandromys oeconomus*), individuals with lower body mass have a better chance of surviving the winter [42]. The condition of an individual of this species has no influence on the dispersal of juveniles and their habitat selection [43]. As confirmed for the short-tailed vole (*Microtus agrestis*) [12,44], the sibling vole (*Microtus rossiaemeridionalis*), *C. glareolus*, and the common shrew (*Sorex araneus*) [45], body condition is involved in the growth or decline of the population. The Chitty effect was not observed in shrews [45].

Due to their small size and high metabolic rate, shrews are more dependent on their environment than rodents, therefore variation in the body condition of shrews may represent an adaptation to climate and habitat productivity [46]. In the case of *S. araneus*, coexistence with the pygmy shrew (*Sorex minutus*) could also influence body size [47]. Regarding the influence of pollution on body condition, in the greater white-toothed shrew (*Crocidura russula*), BCI was not correlated with accumulated metal concentrations, suggesting that BCI is influenced by local environmental factors [28].

Finally, in six species of mice and voles, *A. flavicollis, A. agrarius, C. glareolus. M. arvalis, M. agrestis*, and *A. oeconomus*, it was shown that litter size was related to female body mass to a greater extent than BCI [48]. However, this study only covered a limited variety of habitats, including apple and plum orchards and currant and raspberry plantations.

Utilizing long-term data from snap-trapping of small mammals in Lithuania (Northern Europe), we sought insights into their body condition. After identifying significant factors affecting BCI variation, including decade and season of trapping, habitat, species, age, and gender, we focused our analysis on inter- and intra-species aspects. The aim of the study was to analyze the BCI of small mammals trapped in Lithuania from 1980 to 2023, considering species, age group, gender, and reproductive status. Despite existing literature on mammalian body condition indices and individual fitness, similar studies on small mammals in our latitudes are scarce. This is the first multi-species approach of body condition of small mammals at middle latitudes.

## 2. Materials and Methods

### 2.1. Small Mammal Collection Methods

Small mammals were collected in Lithuania between 1980 and 2023. Snap-trapping was the main method used, with more than 99.9% of specimens collected. In most cases, traps were arranged in lines of 25 traps, 5 m apart, set for three days and checked once or twice a day, i.e., in the morning or in the morning and evening. In most cases, brown bread and crude sunflower oil were used as bait, but in a few trapping sessions the traps were baited with plant material such as a piece of carrot and/or apple. The bait was replaced after rain or when it was eaten.

### 2.2. Small Mammal Trapping and Sample Size

Small mammals were trapped at 321 sampling sites (Figure 1), each representing 1–1950 small mammals. As noted by Balčiauskas and Balčiauskienė [49], equal trapping effort across habitats could not be ensured over the long term. In addition, in the old laboratory protocols related to trapping in fragmented sites, which encompassed multiple habitats, individuals were not always clearly associated with a single habitat. Out of the total sample of 28,567 small mammals, 1181 were trapped in fragmented habitats, including a mix of forest, meadow, wetland, and agricultural land. Of all processed individuals, 7683 were trapped in meadows, 7657 in forests, 4575 in commensal habitats, 2271 in wetlands, 2139 in agricultural habitats, 2022 in disturbed habitats, 647 in riparian habitats, and 392 individuals in shrub habitats. In this article, we do not analyze the influence of habitat, as these results are presented in a separate publication.

The captured small mammals were stored frozen and transferred to the laboratory. Species of small mammals were identified based on their external features. For *Microtus* voles, species were distinguished by differences in their teeth [50]. Genetic methods were employed to identify *M. rossiaemeridionalis* voles.

The eighteen small mammal species (Table 1) represented four trophic groups, mainly omnivores (37.6%) and granivores (34.2%). Herbivores (16.2%) and insectivores (12.0%) were less represented. Among the voles, 53 individuals were not identified by species and were therefore not analyzed further. Two rat species, the brown rat (*Rattus norvegicus*) and the black rat (*Rattus rattus*), were not included due to differences in trapping methods.

The sample was dominated by *C. glareolus* (36.0%), while *A. flavicollis* (19.4%) and *A. agrarius* (13.2%) each accounted for over 10%. The other two species, *M. arvalis* and *S. araneus*, were represented with 8.9% each, *A. oeconomus* with 4.7% of all examined individuals. The proportion of the other species was lower (Table 1).

At the species level, adult animals significantly outnumbered other age groups in the investigated *A. flavicollis* (46.4%) and *A. oeconomus* (51.4%), while non-breeding subadult animals predominated in *S. araneus* (51.9%), and juveniles were most abundant in *A. agrarius* (50.2%) and *M. arvalis* (52.5%). In *C. glareolus*, all three age groups were equally represented (Table 1). Male individuals significantly outnumbered females in samples of *C. glareolus*, *A. flavicollis*, *A. agrarius*, and *S. araneus*. The sex ratio in samples of *M. arvalis* and *A. oeconomus* did not significantly differ from 1:1.

### 2.3. Body Size and Body Condition Index

Captured small mammals were weighed to the nearest 0.1 g. Standard body measurements (body length, tail length, hind foot length, ear length) were recorded to the nearest 0.1 mm using mechanical or electronic calipers. Over 80% of the measurements were conducted by the same person across all study period, therefore minimizing possible bias accordingly [20].

To maintain compatibility with previous research, individual fitness was calculated as the body condition index (BCI) according to Moors [19]: BCI = (Q/L^3^) × 10^5^, where Q is the body weight in grams and L is the body length in millimeters. For pregnant females, the weight of the uterus with embryos was excluded [17].

### 2.4. Estimation of Small Mammal Age and Reproductive Status

Age groups and reproductive status of small mammals were identified under dissection. We divided all small mammals into three age groups: adults, subadults, and juveniles. In the order of importance, age group was defined based on the status of sex organs [52,53,54], then on the presence and atrophy of the thymus gland [55]. Body mass was only used as an indicator of age class when the two previous parameters were unavailable.

All reproductive or post-reproductive individuals were identified as adults, as were those with fully involuted thymus. Based on the size and appearance of the reproductive organs [56,57], we defined five reproductive stages in adult males (Table 2).

In females, we considered as adult all individuals with perforated or plugged vagina, lactating, pregnant, and those with signs of existing pregnancy (embryos) or previous pregnancies, such as *corpora lutea* in the ovaries [58] and/or placental scars in the uterus [59,60].

The number of identified litters was characterized by the largest simultaneous combination of above-listed breeding traits. The two litters may be a combination of signs of an earlier and a more recent litter (embryos and placental scars; presence of old and new placental scars in the uterus) or signs of an earlier pregnancy and fresh signs of conception (open vagina, vaginal plug, presence of sperm in the uterus). As a rule, the three litters indicate the presence of old and new placental scars in the uterus in addition to fresh signs of conception.

Individuals with developed, albeit inactive, genitalia, i.e., without signs of reproductive activity, and partially involuted thymus glands were defined as subadult.

Non-breeding individuals with undeveloped reproductive organs (testes in the abdominal cavity in males, filiform uterus and closed vagina in females) and a well-developed thymus were defined as juveniles [52,53].

### 2.5. Data Analysis

The BCI was characterized using standard statistical parameters: mean, standard error (SE), minimum, and maximum. The normality of the BCI distribution was tested in PAST version 4.13 (Museum of Paleontology, Oslo College, Oslo, Norway) [61]. Since the normality of the BCI distribution could not always be confirmed, we relied on the interpretation of [62], which states that it is still possible to apply the Gaussian generalized linear model (GLM).

We performed GLM analysis with BCI as the dependent parameter, decade, season, habitat, species, individual age, and sex as categorical factors, and body mass and body length as continuous predictors to control for data variability. Outliers were not excluded. Single factor analyzes were performed using ANOVA and differences between groups were analyzed using post-hoc Tukey’s test with unequal sample size [63].

Calculations were performed in Statistica for Windows, version 6.0 (StatSoft, Inc., Tulsa, OK, USA). The minimum confidence level was set as *p* < 0.05.

## 3. Results

All factors examined were significantly related to the variability of BCI, with the model explaining 85.4% of the variance. Among categorical factors, species had the most influence (F_18,25961_ = 1496.6), followed by animal age (F_2,25961_ = 1161.3) and gender (F_2,25961_ = 34.3). The influence of trapping decade (F_4,25961_ = 19.4), season (F_3,25961_ = 13.6), and habitat (F_8,25961_ = 16.5) was less expressed, but all listed categorical factors were significant at *p* < 0.0001. The analysis of the first three factors–species, age, and gender–is presented below.

### 3.1. BCI Differences in Relation to Small Mammal Species

In general, the mean BCI for small mammals in Lithuania was 3.03, with a rather wide range from 1.04 to 6.89. However, the central statistics show small standard errors of the mean for species with a sample size approaching or exceeding 100 individuals (Appendix A).

Small mammals with the highest BCI include *M. avellanarius, A. amphibius, M. minutus*, and *M. musculus*, representing three different trophic groups (Figure 2). The second group with lower BCI contains only granivorous species, *S. betulina, A. agrarius, A. flavicollis*, and *A. sylvaticus*. The third group, with an average value of BCI between 2.82 and 3.0, contains both *Sorex* species and the four vole species of *Alexandromys, Microtus*, and *Clethrionomys*, which in turn represent different trophic groups. The last group with a BCI < 2.82 is also represented by species from three trophic groups (Figure 2), but for all these species the sample size is N < 100.

### 3.2. Age of Individual and Body Condition

We found significant differences in BCI between age groups in six small mammal species (Figure 3). In *S. araneus*, the BCI increases with age, being significantly lower in juveniles than in subadults (post hoc, *p* < 0.01) and lower in subadults than in adults (*p* < 0.001). In four other small mammal species, BCI decreased with the age of the animal, i.e., it was highest in juveniles and lowest in adults. Following this pattern, BCI differed significantly between all age groups in *A. agrarius* (post-hoc, *p* < 0.001) and *C. glareolus* (*p* < 0.02). In *A. flavicollis*, the BCI of juveniles was higher than that of adults and subadults (*p* < 0.005), but not between subadults and adults (Figure 3). In *M. musculus*, the BCI of adults was significantly lower than that of juveniles (post-hoc, *p* < 0.05). In *M. arvalis*, the BCI of juveniles was higher than that of subadults (*p* < 0.01), while the BCI of subadults and adults was similar (Figure 3).

The above patterns were also found in other small mammal species (Appendix A). The decrease in BCI with age was characteristic of *A. uralensis*, *M. minutus*, *A. amphibius*, and *M. agrestis*. An increase in BCI in animals older than juveniles was observed in *S. minutus*, *N. milleri*, and *S. betulina*. In the other species examined, the patterns were different (Appendix A), but, according to the post hoc test, all these differences were not significant.

### 3.3. Gender of Individual and Body Condition

Based on ANOVA analyzes, sex-specific differences in BCI were only found in four small mammal species (Figure 4). Females had a higher BCI than males in *A. agrarius* (post-hoc, *p* < 0.001) and *C. glareolus* (*p* < 0.005), while males had a higher BCI in *S. minutus* (*p* < 0.05). The sample size of *S. betulina* was small, so although a trend was significant (ANOVA, F_1,15_ = 5.74, *p* < 0.05), the higher BCI in males of this species was not confirmed by post-hoc (*p* = 0.062).

In other small mammal species, differences between males and females in BCI were not confirmed by ANOVA and post-hoc (Appendix A). We did not find a correlation with sample size, as the lack of sex-specific differences was also characteristic of abundant small mammal species such as *S. araneus*, *M. musculus*, *A. flavicollis*, *M. minutus*, *A. oeconomus*, *M. agrestis*, and *M. arvalis*.

### 3.4. Influence of the Reproductive Status

Reproductive status had a significant effect on the BCI of *C. glareolus* (F_1,1645_ = 42.1, *p* < 0.0001), *A. flavicollis* (F_1,1532_ = 127.2, *p* < 0.0001), *M. arvalis* (F_1,270_ = 4.3, *p* < 0.05), *A. oeconomus* (F_1,236_ = 8.7, *p* < 0.005), and *S. minutus* (F_1,87_ = 8.4, *p* < 0.005) males, and on the BCI of *M. arvalis* (F_1,451_ = 10.3, *p* < 0.005) and *S. araneus* (F_1,245_ = 13.0, *p* < 0.0005) females (Figure 5). We do not find that these differences are related to sample size, as observed in many other species the BCI of breeding and non-breeding males and females did not differ (Appendix A).

In a number of small mammal species, non-breeding adults had a higher BCI than breeding adults. These differences were significant in males of *A. flavicollis*, *C. glareolus*, and *A. oeconomus* (Figure 5a), whereas the differences were not significant in males of *S. araneus*, *A. agrarius*, and *M. minutus* (Appendix A) and in females of *M. musculus*, *A. agrarius*, *A. uralensis*, *C. glareolus*, *A. oeconomus*, and *M. agrestis* (Appendix A).

However, both males and females of *S. araneus*, *S. minutus*, and *M. arvalis* were characterized by a significantly higher BCI for the breeding individuals (Figure 5a,b). The same pattern, although not significant, was observed in the males of *S. betulina*, *M. musculus*, and *M. agrestis* (Appendix A) and in the females of *S. minutus*, *A. flavicollis*, and *M. musculus* (Appendix A).

As far as the number of litters produced by the female is concerned, more litters mean a lower BCI. This was most pronounced in *C. glareolus* (F_2,621_ = 15.8, *p* < 0.0001), *A. flavicollis* (F_2,425_ = 14.7, *p* < 0.0001), *M. arvalis* (F_2,69_ = 5.3, *p* < 0.01), and *A. agrarius* (F_2,132_ = 4.4, *p* < 0.02). For *A. oeconomus*, the relation was significant (F_1,134_ = 5,1, *p* < 0.05), but a maximum of two litters were recorded in this species (Figure 6). A negative relationship between the number of litters and BCI was also found in *M. musculus*, *M. minutus*, and *M. agrestis* (Appendix A), but there were only a maximum of two litters and the sample size was small, N < 50.

In males, the predominant pattern shows a decrease in BCI along with the progression of the intensity of spermatogenesis, although there are some variations (Figure 7 and Appendix A). This decrease was strong and significant in *C. glareolus* (F_4,1192_ = 11.3, *p* < 0.0001), similar in *S. minutus* (F_2,12_ = 10.3, *p* < 0.005), although some breeding periods were not observed, and pronounced but not significant in *A. oeconomus*. A similar pattern was less pronounced in *A. agrarius*, *M. agrestis*, and *M. rossiaemeridionalis*. In some species, BCI increased again in males after the breeding season, as in *A. flavicollis* (F_4,817_ = 4.2, *p* < 0.005) and *M. minutus* (NS). In contrast, BCI increased in *M. arvalis* as the breeding season progressed (F_4,179_ = 4.2, *p* < 0.005), but decreased thereafter (post-hoc, *p* < 0.05).

## 4. Discussion

Small mammal BCI in Lithuania exhibits a considerable variation, with a range exceeding sixfold. Our findings revealed that eighteen species form four clusters with different BCIs, three of which include species from different trophic groups. Interestingly, the group comprising solely granivorous species exhibits an above-average BCI, indicating that the average species BCI and trophic group are not directly related. We found that higher BCIs are characteristic of granivorous and omnivorous species, which may switch and at least seasonally use animal foods. This is confirmed to the seven out of eight species with highest BCI: *M. avellanarius* [64,65], *M. minutus* [66,67], *M. musculus* [68], *S. betulina* [69], *A. agrarius* [70], *A. flavicollis* [71], and *A. sylvaticus* [72,73]. Among these species, only *A. amphibius* is a typical herbivore [74].

We found two contrasting patterns of age-related BCI dynamics. During ontogeny, BCI increased in *S. araneus*, *S. minutus*, *N. milleri*, and *S. betulina*, but not significantly in the latter three species. In *A. agrarius*, *C. glareolus*, *A. flavicollis*, *M. musculus*, *M. arvalis*, *A. uralensis*, *M. minutus*, *A. amphibius*, and *M. agrestis*, the BCI decreased (see Figure 3 and Appendix A). The first pattern means that body mass grows faster than body size. According to G.L. Blackwell [21], the BCI could be unequal due to differences in diet, as it increases when more nitrogen-containing food is eaten. This confirmed the opinion of T.C.R. White [75] on the importance of nitrogenous food. In a previous study, we confirmed that the BCI of *A. flavicollis* is high in the cormorant colony area where the rodents’ diet was enriched with nitrogen [30].

When BCI is compared between age groups of the same species, the growth effect in the early life phase could have an influence [76], therefore the correlations between body mass and body size should be similar.

We tested correlations between body mass and body size in different age groups of each species (Appendix A) and found that these are least pronounced in shrews. The other conspicuous trait, body mass and size, correlated least in subadult animals, i.e., in the non-breeding stage. However, a detailed analysis of these patterns was not one of the aims of this study.

It should be noted that differences in body mass may vary in response to different ecological factors. Similar differences have been confirmed for three Korean rodent species: the red-backed vole (*Myodes regulus*), the Korean field mouse (*Apodemus peninsulae*), and *A. agrarius* [77]. These authors do not analyze the body condition index, but their study shows species-specific ecological factors that influence body mass. One can consider other factors than BCI that influence small mammal fitness, such as pollution, genetic factors, behavioral factors, health and diseases, etc. However, we have no information about these factors in our study.

The second factor for the age-related difference in BCI could be the reproductive stress of the adult animals. Our data fully confirmed the following assumption for females of all species (see Figure 6 and Appendix A). In males, the decrease in BCI with the progression of reproduction was characteristic of most species, with few exceptions to this pattern (Figure 7 and Appendix A). Our results therefore fully confirm the statement by J.R. Speakman [78] that the cost of reproduction in females is due to the “increased energy, protein and calcium demands during pregnancy, but most particularly during lactation”.

Differences in the BCI patterns of shrews should be considered as an adaptation to their small body size and high energy expenditure (Frafjord, 2008). Therefore, shrew species may respond differently to environmental factors compared to rodents [47]. Their smaller size in winter significantly reduces their energy requirements and thus increases their chances of survival [79]. In our study, of the 26 small mammals whose body condition was half the average, i.e., BCI < 1.50, 46.2% were *S. minutus* and 26.9% were *S. araneus*. Moreover, most of these shrews were caught in late fall and winter.

In contrast, there are very large individuals that are characterized by an increased body mass. In *M. californicus*, extra-large individuals accounted for 12.7% of the total sample [34]. Very large individuals were recorded in *M. agrestis, M. rossiaemeridionalis*, and *C. glareolus* [80] but not in shrews [45]. In live-trapped shrews, body mass decreases with time spent in traps [81], so data may not be comparable if the trapping method is not clearly stated. We had only a few specimens that were not caught with snap traps, so the BCI of shrews in our sample is fully comparable. It is important to note that the body condition index is only a proxy for individual fitness. As a result, BCI value in explaining the Chitty effect may be limited.

In our sample, there were 141 individuals with a BCI > 5, which corresponds to almost 0.5% of the total number. Among these, the dominating species were *A. flavicollis* (20.6%) and *M. minutus* (17.0%). Very large *A. agrarius* accounted for 13.5%, *C. glareolus* for 10.6%; *M. musculus* and *A. oeconomus* accounted for 8.5% each. We also recorded very large shrews, *S. araneus* with 8.5% and *S. minutus* with 7.8% of all individuals with BCI > 5. The proportion of large herbivores, *M. arvalis*, *M. agrestis*, and *A. amphibius* was lower.

In mammals in general, mortality may serve as the best predictor of variation in life history, but only after accounting for the effects of body weight, which is analogous to BCI [82]. The BCI of primitive insectivores might be expected to differ from that of herbivores, as the latter group generally has more abundant and reliable food sources [17]. With the exception of a single *M. avellanarius*, our sample does not include hibernating small mammals, which would exhibit other age-specific differences and adaptations to the temperature regime [83].

What else is important about BCI studies in the context of climate change? Better individual fitness could lead to higher chances of surviving the winter. This has been shown for *A. flavicollis* [84], *M. agrestis* [6,12], and *C. glareolus* [40,41]. However, when non-breeding animals stop growing, a high BCI can be maintained with a lower body mass, as is the case with *A. oeconomus* [42].

The effects of climate change on vegetation are also important. In the northern hemisphere, there is a trend towards a longer growing season in grasslands [85]. The warming in spring that has been observed over the last decade will also lengthen growing seasons [86]. Therefore, a short winter, a long growing season, and warmer average temperatures will have a positive effect on the body condition of animals, including rodents [83]. Given the increased chances of survival [44], we can expect an increase in the number of small mammals. Survival of larger individuals will be associated with higher reproductive parameters [48] and could have a positive effect on increased ectoparasite burden [36].

## 5. Conclusions

This is the first multi-species investigation of the body condition index of small mammals in the middle latitudes that compares BCI between species and analyzes differences between age groups, sex, and reproductive status of males and females within species.

At the species level, we conclude that of the eight species with the highest average BCI, seven are granivores or omnivores that use food of animal origin at least seasonally. Two contrasting age-related BCI patterns were found: decreasing or increasing during ontogeny. We hypothesize that these patterns may be related to differences in the diets of juveniles, subadults, and adults. Our results demonstrate a negative influence of reproductive stress on the BCI of all adult females and nearly all adult males. Among animals with exceptionally low BCI, shrews were predominant. Notably, we observed the Chitty effect in *S. araneus* and *S. minutus*, with these species exhibiting very large body mass resulting in a BCI > 5.0.

The results increase our understanding of how changing environmental conditions are affecting small mammals. To deepen this understanding, we intend to analyze body condition indices in the most abundant species, considering habitat dynamics and adopting a long-term perspective.

Our study is limited by the lack of pathogen load data from older studies and a lack of knowledge about diet. The latter is available in the form of nitrogen and carbon isotope concentrations in small mammals from commercial orchard material, which can be analyzed in the future.

## Figures and Tables

**Figure 1 animals-14-01686-f001:**
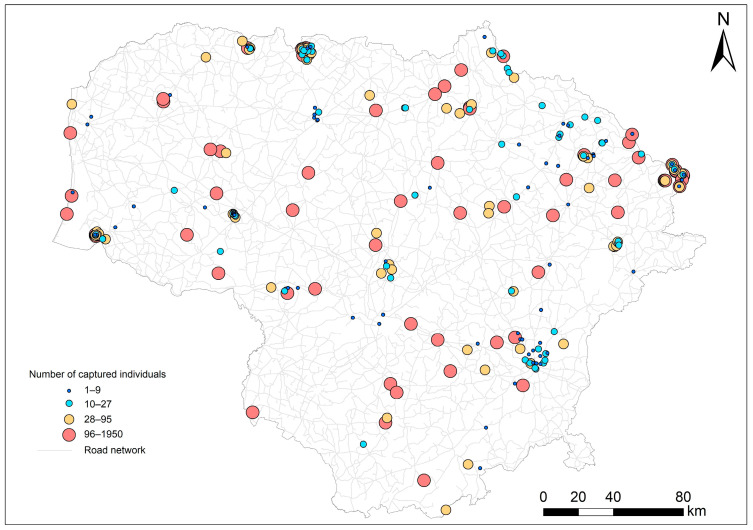
Small mammal trapping sites in Lithuania, 1980–2023. Dot size corresponds to the number of analyzed individuals.

**Figure 2 animals-14-01686-f002:**
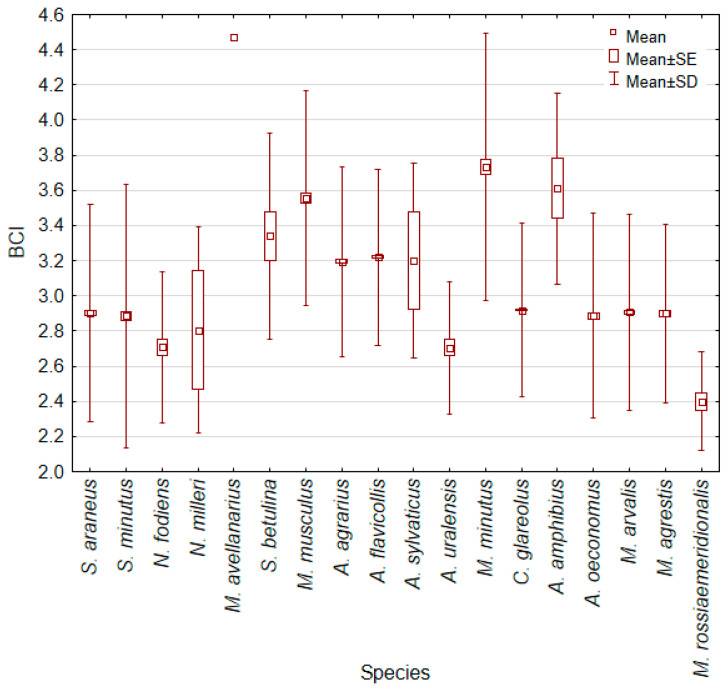
Body condition index (BCI) of small mammals independent of their age and sex.

**Figure 3 animals-14-01686-f003:**
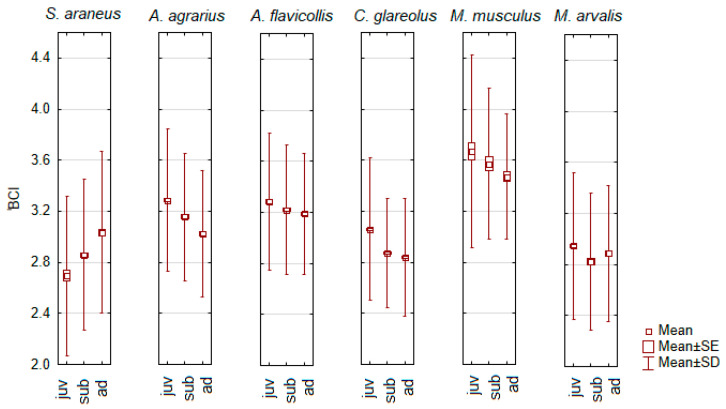
Significant age-related differences in the body condition index (BCI) of small mammals.

**Figure 4 animals-14-01686-f004:**
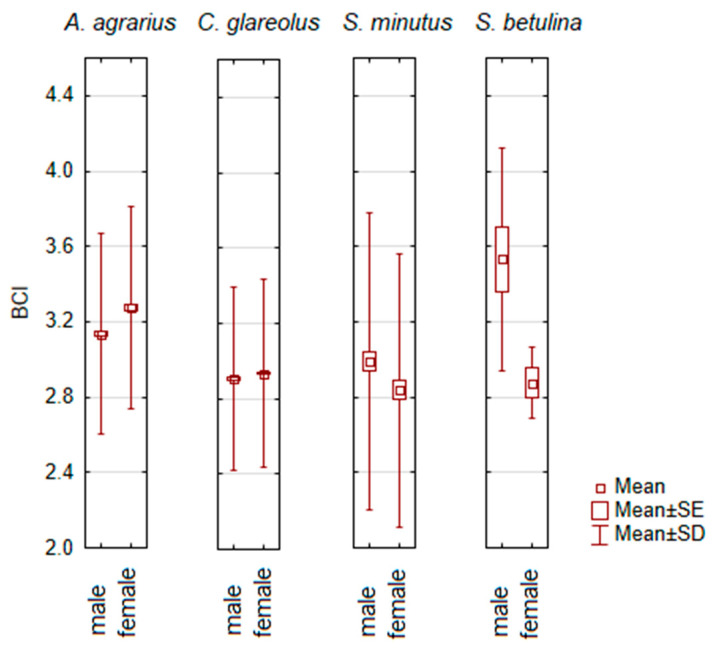
Significant sex-related differences in body condition index (BCI) of small mammals.

**Figure 5 animals-14-01686-f005:**
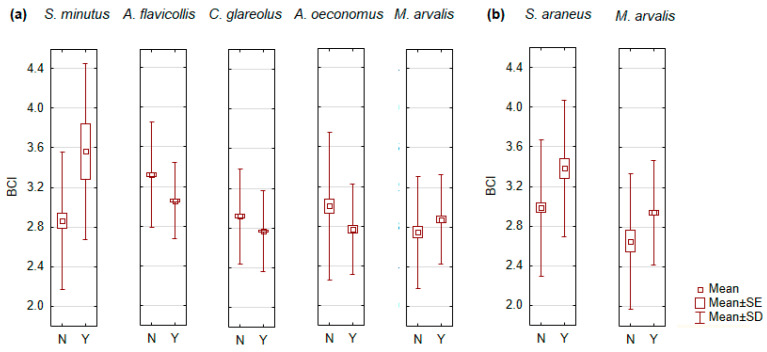
Significant reproduction-related differences in the body condition index (BCI) of males (**a**) and females (**b**) of small mammals: N–non-breeding, Y–breeding individuals.

**Figure 6 animals-14-01686-f006:**
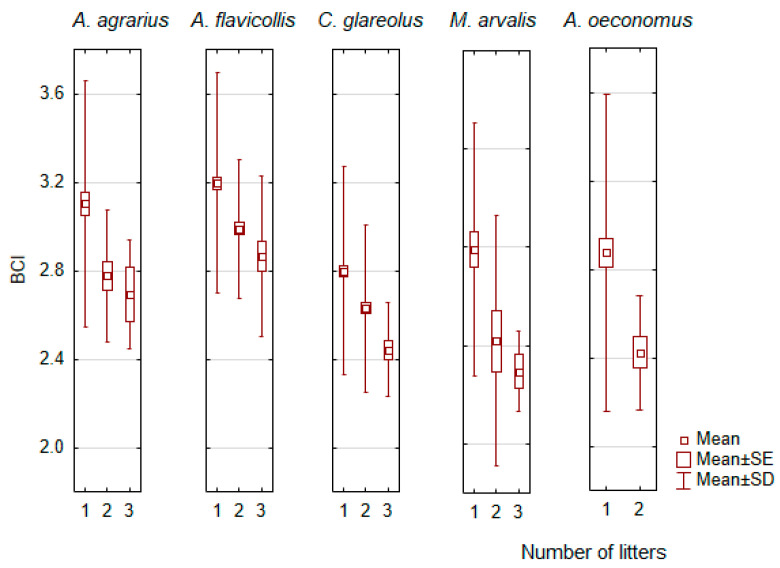
Significant differences in body condition index (BCI) depending on the number of litters produced by the females.

**Figure 7 animals-14-01686-f007:**
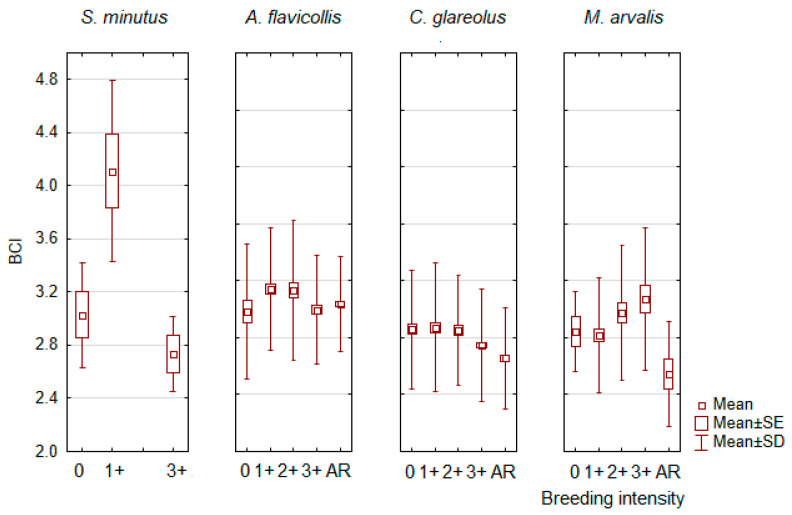
Significant differences in body condition index (BCI) in relation to breeding intensity in males: 0–no spermatogenesis, 1+–beginning of spermatogenesis, 2+–spermatogenesis of medium intensity, 3+–intensive spermatogenesis, AR–individuals after reproduction.

**Table 1 animals-14-01686-t001:** Sample composition of small mammals Trophic groups (TG): I—insectivores, G—granivores, H—herbivores, O—omnivores, according to [49,51]. Age groups: Ad–adult, Sub–non-breeding, Juv–juvenile animals.

Species	TG	N *	Male **	Female	Ad	Sub	Juv
Common shrew (*Sorex araneus*)	I	2536	1079	739	658	1315	270
Pygmy shrew (*S. minutus*)	I	805	260	202	159	128	35
Water shrew (*Neomys fodiens*)	I	100	44	33	48	38	7
Mediterranean water shrew (*N. milleri*)	I	3	1	2	1		1
Hazel dormouse (*Muscardinus avellanarius*)	O	1		1			1
Northern birch mouse (*Sicista betulina*)	G	18	12	6	11	3	4
House mouse (*Mus musculus*)	O	432	271	161	183	117	132
Striped field mouse (*Apodemus agrarius*)	G	3781	2152	1580	774	1067	1898
Yellow-necked mouse (*A. flavicollis*)	G	5561	2912	2566	2583	1621	1284
Wood mouse (*A. sylvaticus*)	G	5	4	1	2	1	2
Pygmy field mouse (*A. uralensis*)	G	74	27	46	20	23	31
Harvest mouse (*Micromys minutus*)	G	347	186	149	58	95	190
Bank vole (*Clethrionomys glareolus*)	O	10,316	5373	4860	3482	3505	3270
Water vole (*Arvicola amphibius*)	H	10	7	3	1	4	5
Root vole (*Alexandromys oeconomus*)	H	1337	622	706	688	238	405
Common vole (*Microtus arvalis*) ***	H	2537	1257	1257	758	427	1333
Short-tailed vole (*M. agrestis*)	H	674	346	326	308	188	177
Sibling vole (*M. rossiaemeridionalis*)	H	30	13	17	19	6	5

*—Differences between the total number of individuals captured (N) and the sum of identified age and gender numbers occurred due to damage to the body, preventing accurate assessment of these characteristics. This issue was most pronounced in shrews; **—One individual each of *A. flavicollis*, *M. agrestis*, and two individuals of *A. oeconomus* were hermaphrodites, exhibiting developed gonads of both sexes; ***—*Sensu lato*. In most studies, *M. rossiaemeridionalis* was not specifically identified.

**Table 2 animals-14-01686-t002:** Description of reproductive stages in adult males.

Stage	Abbreviation	Description
No spermatogenesis	(0)	testes scrotal, but no other signs, thymus absent
Beginning of spermatogenesis	(1+)	testes enlarged, epididymis empty
Spermatogenesis of medium intensity	(2+)	testes and epididymis enlarged, seminal vesicles empty or slightly enlarged
Intensive spermatogenesis	(3+)	testes enlarged, epididymis and seminal vesicles full
Individuals after reproduction	(AR)	seminal vesicles empty, testes and epididymis slumped, slate colored

## Data Availability

This is ongoing research, therefore data are available from the corresponding author upon request.

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
