# Peer review of "Insight into Body Condition Variability in Small Mammals"

_animals, 2024, doi:10.3390/ani14111686_

Round 1

Reviewer 1 Report

Comments and Suggestions for Authors

This paper discusses multi-species investigation of the body condition index of small mammals in the middle latitudes, that compares body condition index between species and analyzes differences between age groups, sex, and reproductive status of males and females within species.  It is really well written and was a pleasure to read. No further comments. 

Author Response

thank you for your kind evaluation!

Reviewer 2 Report

Comments and Suggestions for Authors

Simple Summary

Sample Size: While 30,000 individuals sound like a large sample, it's important to consider the number per species. The authors mention 18 species, but don't specify sample size per species. Low sample sizes within a species could weaken the findings.

Habitat Variation: The study covers a 43-year period. Lithuania's habitat might have changed over time. The authors should address if habitat variations were considered and if they might affect BCI.

Seasonal Variation: BCI can fluctuate seasonally. The study doesn't mention if trapping was conducted throughout the year or just specific seasons. This could affect comparisons between species.

Cause and Effect: The study finds a correlation between reproductive stress and lower BCI, but doesn't establish causation. Other factors like disease could be influencing BCI.

Chitty Effect: The "Chitty effect" explanation for high BCI in shrews needs elaboration. Are there specific mechanisms or data to support this claim?

Sentence 11: "granivore or omnivore species with the highest average BCI consume animal food at least occasionally" - This sentence could be rephrased for clarity: "Among the species with the highest average BCI, seven out of eight granivores or omnivores consume animal food occasionally."

Sentence 16: "We observed a rare case in two species of shrews" - Consider specifying "the Chitty effect" here for better flow.

Abstract:

Sample size per species is still not mentioned.

Habitat variation and potential impact are not addressed.

"BCI...indicates their reproductive success and health" - Consider rephrasing for clarity: "BCI is an indicator of both reproductive success and health in small mammals."

"representing 18 small mammal species, which were trapped" - Replace "which" with "that" for better flow.

"considering their species, age group, gender and reproductive status" - "Considering" sounds awkward here. "We analyzed BCI...considering..." is better.

"food of animal origin at least seasonally" - Consider "animal-based food" for readability.

"extremely low BCI were mostly shrews" - "Consisted mostly of shrews" or "Shrews were dominant among..."

Introduction

The introduction effectively discusses the importance of BCI and various factors affecting it, but it could be strengthened by better highlighting the novelty of the study: investigating BCI in small mammals of middle latitudes.

While the references cited discuss BCI in various species, the focus on middle latitudes isn't emphasized in this section.  More emphasis is needed on how this study fills the gap in knowledge about BCI specifically in small mammals of mid-latitudes.

The introduction now explicitly mentions the lack of BCI studies on small mammals in middle latitudes (sentence 64). This is a good step towards establishing the novelty of the research.

However, it could be further strengthened by acknowledging the limitations of BCI as a measure of fitness, especially when discussing dispersal and the Chitty effect.

The introduction would be even stronger by explicitly stating the research question.

Consider mentioning the limitations of using BCI as a proxy for fitness. Briefly acknowledge that other factors can influence fitness besides body condition.

Briefly mention the study methods planned to address the research question.

L41: Replace "BMI" with "BCI" (Body Condition Index) as BMI is used for humans.

L50: Consider rephrasing "The Dehnel phenomenon... explains..." to "Small shrews survive winter by..." for a smoother flow.

L52: "Voles increase chances of survival..." - Consider "by accumulating fat reserves..." to clarify the strategy.

L57: "Body condition may play a role..." - "May" could be strengthened to "likely plays" since the introduction has established the importance of BCI.

L68: "Various factors were analyzed..." - Consider "Several studies have analyzed..." for a more natural flow.

L75: "Schweiger et al. [27] caution..." - Consider "Some researchers caution..." for better flow.

L92: "Other references are..." - This sentence can be omitted because the focus has shifted to the study itself.

L93: "yellow-necked mouse" - This is the first mention. Add the full scientific name.

Finally, the introduction becomes long and boring. Please keep it short and focus on your topic. In fact, the introduction is more like a discussion than an introduction.

Method

Habitat is mentioned as a factor influencing BCI in the introduction, but then excluded from the analysis here (sentence 155). This is a significant decision and needs justification.

The authors acknowledge unequal trapping effort across habitats (sentence 147). This could bias the results towards habitats with more trapping effort.

The total number of individuals trapped in fragmented habitats is mentioned (1181), but the distribution across forest, meadow etc. within fragmented habitats is not provided.

The total number of individuals trapped (28,620) doesn't match the sum of individuals listed by specific habitat types (total = 26,442). There's a discrepancy of 2,178 individuals for unaccounted habitats.

The exclusion of some species (brown rat, black rat, and unidentified voles) is mentioned (sentence 166) but not justified. Were there health concerns, identification issues, or other reasons?

While both balance and electronic scales are mentioned (sentence 192), it would be good to clarify if there were any systematic differences in measurements between the two types.

The results show contrasting patterns of BCI and age across species. While some species show increasing BCI with age, others show a decrease. The text acknowledges this variation but doesn't provide a strong explanation for the differences (sentences 278-279).

The results acknowledge that the sex-specific BCI difference observed in S. betulina might be unreliable due to a small sample size (sentence 298).

The results show contrasting patterns of BCI and reproductive status across species. For example, breeding can increase or decrease BCI depending on the species (Figures 5, 6, 7). The text acknowledges this variation but could benefit from a more thorough discussion of the potential biological mechanisms (e.g., energetic demands of reproduction) underlying these differences.

L146: "Material was collected..." - Consider "Small mammals were trapped..." for a clearer start.

L148: "Due to several decades-old laboratory protocols..." - This justification for unequal trapping effort could be explained more concisely.

151: "fragmented habitats, including forests, meadows, wetlands, and agricul- 152 tural lands" - This could be rephrased: "including a mix of forest, meadow, wetland and agricultural land."

L160: "The captured individuals..." - Consider "Small mammals..." for better flow.

L192: "balance, spring" - Consider replacing with "balance scales, spring scales" for clarity.

L193: "standard measures... were taken...to the nearest 0.1 mm" - This could be rephrased for conciseness: "Standard body measurements (mass, length, tail length, hind foot length, ear length) were recorded to the nearest 0.1 mm using mechanical or electronic calipers."

Consider "Over 80% of the measurements were conducted by the same person..."

L203: "The age groups and reproductive status..." - Consider "Age groups and reproductive status..."

L206: "Body mass as an indicator..." - This could be rephrased: "Body mass was only used as an indicator of age class when the two previous parameters were unavailable."

L 210-216: Consider combining these sentences into a table for clarity

L  223-229: This section describing how litters were identified could be condensed and rephrased for improved readability.

L  241: "Gaussian family GLM..." - Consider "Gaussian generalized linear model (GLM)..." for brevity.

Results

The results acknowledge that M. avellanarius and A. amphibius (species with high BCI) are underrepresented in the sample (sentence 265). While they are excluded from further analysis, consider briefly discussing the potential consequences of this exclusion on the overall findings.

L253: "All tested factors..." - This could be rephrased for better flow: "All factors examined were significantly related..."

L261: "rather wide range" - Consider replacing with a more specific descriptive statistic, e.g., "standard deviation".

L266: "analyzed further" - Consider replacing with "included in further analysis".

L269: Consider rephrasing "2.82< BCI < 3.0" to "BCI between 2.82 and 3.0".

L276: "We found significant differences..." - This could be rephrased for conciseness: "Significant differences in BCI..."

L280: "all differences between age groups..." - Consider "BCI differed significantly between all age groups..."

L283: Consider rephrasing "although the BCI was similar in the latter two age categories" to "but not between subadults and adults".

Figure references: Double-check the numbering of supplementary figures (Figure S1 mentioned in sentence 286, but Figure 2 mentioned in sentence 292).

L309: "Participation in the reproductive processes..." - This could be rephrased for conciseness: "Reproductive status..."

L314: "as in numerous other species..." - Consider replacing with "as observed in many other species" for better flow.

Ensure consistent numbering for supplementary figures (e.g., Figure S3 mentioned multiple times).

Discussion

The discussion acknowledges contrasting patterns of BCI across species (e.g., granivores vs shrews, Figure 3) but could benefit from a more in-depth discussion of the underlying biological reasons.

The discussion cites several studies to support findings, however, it might be beneficial to broaden the scope by incorporating a wider range of relevant literature on BCI and ecological factors.

L 352: "The average BCI..." - This could be rephrased for improved flow: "Small mammal BCI in Lithuania exhibits..."

L 360: "Only one of these species, A. amphibius, is typical herbivore..." - Consider "Only A. amphibius among these species is a typical herbivore..."

L 373: "When comparing the BCI..." - This sentence could be restructured for clarity: "When BCI is compared between age groups..."

L 382: "such differences..." - Consider replacing with "similar differences" for better coherence with the previous sentence.

L 400: "On the other hand..." - This could be replaced with a transition indicating a contrasting point, like "In contrast..."

Conclusions

No limitations of the study are mentioned.

The conclusion could benefit from a stronger statement about the novelty of the findings, especially regarding the multi-species approach at mid-latitudes (mentioned in the introduction).

Please remove the conclusion from the state of cutting the sentences and let the sentences continue.

Comments on the Quality of English Language

Simple Summary

Sample Size: While 30,000 individuals sound like a large sample, it's important to consider the number per species. The authors mention 18 species, but don't specify sample size per species. Low sample sizes within a species could weaken the findings.

Habitat Variation: The study covers a 43-year period. Lithuania's habitat might have changed over time. The authors should address if habitat variations were considered and if they might affect BCI.

Seasonal Variation: BCI can fluctuate seasonally. The study doesn't mention if trapping was conducted throughout the year or just specific seasons. This could affect comparisons between species.

Cause and Effect: The study finds a correlation between reproductive stress and lower BCI, but doesn't establish causation. Other factors like disease could be influencing BCI.

Chitty Effect: The "Chitty effect" explanation for high BCI in shrews needs elaboration. Are there specific mechanisms or data to support this claim?

Sentence 11: "granivore or omnivore species with the highest average BCI consume animal food at least occasionally" - This sentence could be rephrased for clarity: "Among the species with the highest average BCI, seven out of eight granivores or omnivores consume animal food occasionally."

Sentence 16: "We observed a rare case in two species of shrews" - Consider specifying "the Chitty effect" here for better flow.

Abstract:

Sample size per species is still not mentioned.

Habitat variation and potential impact are not addressed.

"BCI...indicates their reproductive success and health" - Consider rephrasing for clarity: "BCI is an indicator of both reproductive success and health in small mammals."

"representing 18 small mammal species, which were trapped" - Replace "which" with "that" for better flow.

"considering their species, age group, gender and reproductive status" - "Considering" sounds awkward here. "We analyzed BCI...considering..." is better.

"food of animal origin at least seasonally" - Consider "animal-based food" for readability.

"extremely low BCI were mostly shrews" - "Consisted mostly of shrews" or "Shrews were dominant among..."

Introduction

The introduction effectively discusses the importance of BCI and various factors affecting it, but it could be strengthened by better highlighting the novelty of the study: investigating BCI in small mammals of middle latitudes.

While the references cited discuss BCI in various species, the focus on middle latitudes isn't emphasized in this section.  More emphasis is needed on how this study fills the gap in knowledge about BCI specifically in small mammals of mid-latitudes.

The introduction now explicitly mentions the lack of BCI studies on small mammals in middle latitudes (sentence 64). This is a good step towards establishing the novelty of the research.

However, it could be further strengthened by acknowledging the limitations of BCI as a measure of fitness, especially when discussing dispersal and the Chitty effect.

The introduction would be even stronger by explicitly stating the research question.

Consider mentioning the limitations of using BCI as a proxy for fitness. Briefly acknowledge that other factors can influence fitness besides body condition.

Briefly mention the study methods planned to address the research question.

L41: Replace "BMI" with "BCI" (Body Condition Index) as BMI is used for humans.

L50: Consider rephrasing "The Dehnel phenomenon... explains..." to "Small shrews survive winter by..." for a smoother flow.

L52: "Voles increase chances of survival..." - Consider "by accumulating fat reserves..." to clarify the strategy.

L57: "Body condition may play a role..." - "May" could be strengthened to "likely plays" since the introduction has established the importance of BCI.

L68: "Various factors were analyzed..." - Consider "Several studies have analyzed..." for a more natural flow.

L75: "Schweiger et al. [27] caution..." - Consider "Some researchers caution..." for better flow.

L92: "Other references are..." - This sentence can be omitted because the focus has shifted to the study itself.

L93: "yellow-necked mouse" - This is the first mention. Add the full scientific name.

Finally, the introduction becomes long and boring. Please keep it short and focus on your topic. In fact, the introduction is more like a discussion than an introduction.

Method

Habitat is mentioned as a factor influencing BCI in the introduction, but then excluded from the analysis here (sentence 155). This is a significant decision and needs justification.

The authors acknowledge unequal trapping effort across habitats (sentence 147). This could bias the results towards habitats with more trapping effort.

The total number of individuals trapped in fragmented habitats is mentioned (1181), but the distribution across forest, meadow etc. within fragmented habitats is not provided.

The total number of individuals trapped (28,620) doesn't match the sum of individuals listed by specific habitat types (total = 26,442). There's a discrepancy of 2,178 individuals for unaccounted habitats.

The exclusion of some species (brown rat, black rat, and unidentified voles) is mentioned (sentence 166) but not justified. Were there health concerns, identification issues, or other reasons?

While both balance and electronic scales are mentioned (sentence 192), it would be good to clarify if there were any systematic differences in measurements between the two types.

The results show contrasting patterns of BCI and age across species. While some species show increasing BCI with age, others show a decrease. The text acknowledges this variation but doesn't provide a strong explanation for the differences (sentences 278-279).

The results acknowledge that the sex-specific BCI difference observed in S. betulina might be unreliable due to a small sample size (sentence 298).

The results show contrasting patterns of BCI and reproductive status across species. For example, breeding can increase or decrease BCI depending on the species (Figures 5, 6, 7). The text acknowledges this variation but could benefit from a more thorough discussion of the potential biological mechanisms (e.g., energetic demands of reproduction) underlying these differences.

L146: "Material was collected..." - Consider "Small mammals were trapped..." for a clearer start.

L148: "Due to several decades-old laboratory protocols..." - This justification for unequal trapping effort could be explained more concisely.

151: "fragmented habitats, including forests, meadows, wetlands, and agricul- 152 tural lands" - This could be rephrased: "including a mix of forest, meadow, wetland and agricultural land."

L160: "The captured individuals..." - Consider "Small mammals..." for better flow.

L192: "balance, spring" - Consider replacing with "balance scales, spring scales" for clarity.

L193: "standard measures... were taken...to the nearest 0.1 mm" - This could be rephrased for conciseness: "Standard body measurements (mass, length, tail length, hind foot length, ear length) were recorded to the nearest 0.1 mm using mechanical or electronic calipers."

Consider "Over 80% of the measurements were conducted by the same person..."

L203: "The age groups and reproductive status..." - Consider "Age groups and reproductive status..."

L206: "Body mass as an indicator..." - This could be rephrased: "Body mass was only used as an indicator of age class when the two previous parameters were unavailable."

L 210-216: Consider combining these sentences into a table for clarity

L  223-229: This section describing how litters were identified could be condensed and rephrased for improved readability.

L  241: "Gaussian family GLM..." - Consider "Gaussian generalized linear model (GLM)..." for brevity.

Results

The results acknowledge that M. avellanarius and A. amphibius (species with high BCI) are underrepresented in the sample (sentence 265). While they are excluded from further analysis, consider briefly discussing the potential consequences of this exclusion on the overall findings.

L253: "All tested factors..." - This could be rephrased for better flow: "All factors examined were significantly related..."

L261: "rather wide range" - Consider replacing with a more specific descriptive statistic, e.g., "standard deviation".

L266: "analyzed further" - Consider replacing with "included in further analysis".

L269: Consider rephrasing "2.82< BCI < 3.0" to "BCI between 2.82 and 3.0".

L276: "We found significant differences..." - This could be rephrased for conciseness: "Significant differences in BCI..."

L280: "all differences between age groups..." - Consider "BCI differed significantly between all age groups..."

L283: Consider rephrasing "although the BCI was similar in the latter two age categories" to "but not between subadults and adults".

Figure references: Double-check the numbering of supplementary figures (Figure S1 mentioned in sentence 286, but Figure 2 mentioned in sentence 292).

L309: "Participation in the reproductive processes..." - This could be rephrased for conciseness: "Reproductive status..."

L314: "as in numerous other species..." - Consider replacing with "as observed in many other species" for better flow.

Ensure consistent numbering for supplementary figures (e.g., Figure S3 mentioned multiple times).

Discussion

The discussion acknowledges contrasting patterns of BCI across species (e.g., granivores vs shrews, Figure 3) but could benefit from a more in-depth discussion of the underlying biological reasons.

The discussion cites several studies to support findings, however, it might be beneficial to broaden the scope by incorporating a wider range of relevant literature on BCI and ecological factors.

L 352: "The average BCI..." - This could be rephrased for improved flow: "Small mammal BCI in Lithuania exhibits..."

L 360: "Only one of these species, A. amphibius, is typical herbivore..." - Consider "Only A. amphibius among these species is a typical herbivore..."

L 373: "When comparing the BCI..." - This sentence could be restructured for clarity: "When BCI is compared between age groups..."

L 382: "such differences..." - Consider replacing with "similar differences" for better coherence with the previous sentence.

L 400: "On the other hand..." - This could be replaced with a transition indicating a contrasting point, like "In contrast..."

Conclusions

No limitations of the study are mentioned.

The conclusion could benefit from a stronger statement about the novelty of the findings, especially regarding the multi-species approach at mid-latitudes (mentioned in the introduction).

Please remove the conclusion from the state of cutting the sentences and let the sentences continue.

Author Response

Rev2 round 1 comments and answers

Thank you very much for detailed comments. We did our best to answer them. Please find revision of the manuscript and answers in separate file. Ther best answer in general is that we expect to have series of papers, and this is the first one. Too many factors will make a mess in presenting, especially to figure and tables, therefore we just list significant factors related to BCI, and then analyze some of them in one paper.

All your comments are very appreciated. Hope you find answers acceptable. Some comments are acknowledged with reserve, as there is very limited number of text allowed in Simple summary and in Abstract.

Simple Summary

Comment: Sample Size: While 30,000 individuals sound like a large sample, it's important to consider the number per species. The authors mention 18 species, but don't specify sample size per species. Low sample sizes within a species could weaken the findings.

Answer: it is not possible to show sample size for all species, so we just add text mentioning that in seven species sample size was small, N < 100.

Comment: Habitat Variation: The study covers a 43-year period. Lithuania's habitat might have changed over time. The authors should address if habitat variations were considered and if they might affect BCI.

Answer: as it was said in the first chapter of Results, habitat had influence, but it was not analyzed further in this manuscript. We already presented next paper with the analysis of habitat to different journal. However, there are no available data on the specific changes of the habitat from the small mammal aspect of view.

Comment: Seasonal Variation: BCI can fluctuate seasonally. The study doesn't mention if trapping was conducted throughout the year or just specific seasons. This could affect comparisons between species.

Answer: as it was said in the first chapter of Results, time factor (both decade and season) had influence, but it was not analyzed further in this manuscript. This will be subject for the third publication we are preparing.

Comment: Cause and Effect: The study finds a correlation between reproductive stress and lower BCI, but doesn't establish causation. Other factors like disease could be influencing BCI.

Answer: we agree with you, but other factors were not analyzed, therefore we cannot relate pathogen presence to BCI. Added as ‘limitation of the study’ as the end of Conclusions.

Comment: Chitty Effect: The "Chitty effect" explanation for high BCI in shrews needs elaboration. Are there specific mechanisms or data to support this claim?

Answer: as for data, we added text ‘Notably, we observed the Chitty effect in S. araneus and S. minutus (23 out of 3341 individuals), with these species exhibiting very large body mass resulting in a BCI > 5.0.’ Mechanism was not investigated, though.

Comment: Sentence 11: "granivore or omnivore species with the highest average BCI consume animal food at least occasionally" - This sentence could be rephrased for clarity: "Among the species with the highest average BCI, seven out of eight granivores or omnivores consume animal food occasionally."

Answer: accepted as suggested.

Comment: Sentence 16: "We observed a rare case in two species of shrews" - Consider specifying "the Chitty effect" here for better flow.

Answer: accepted as suggested.

Abstract:

Comment: Sample size per species is still not mentioned.

Answer: Inserted that sample size for seven species was < 100-

Comment: Habitat variation and potential impact are not addressed.

Answer: apologies, we did not find possibility to fit this text onto 200 word limit.

Comment: "BCI...indicates their reproductive success and health" - Consider rephrasing for clarity: "BCI is an indicator of both reproductive success and health in small mammals."

Answer: accepted as suggested.

Comment: "representing 18 small mammal species, which were trapped" - Replace "which" with "that" for better flow.

Answer: accepted as suggested.

Comment: "considering their species, age group, gender and reproductive status" - "Considering" sounds awkward here. "We analyzed BCI...considering..." is better.

Answer: sentence changed

Comment: "food of animal origin at least seasonally" - Consider "animal-based food" for readability.

Answer: accepted as suggested.

Comment: "extremely low BCI were mostly shrews" - "Consisted mostly of shrews" or "Shrews were dominant among..."

Answer: first suggestion accepted.

Introduction

Comment: The introduction effectively discusses the importance of BCI and various factors affecting it, but it could be strengthened by better highlighting the novelty of the study: investigating BCI in small mammals of middle latitudes.

Answer: comment acknowledged.

Comment: While the references cited discuss BCI in various species, the focus on middle latitudes isn't emphasized in this section.  More emphasis is needed on how this study fills the gap in knowledge about BCI specifically in small mammals of mid-latitudes.

Answer: we added explaining sentence.

Comment: The introduction now explicitly mentions the lack of BCI studies on small mammals in middle latitudes (sentence 64). This is a good step towards establishing the novelty of the research.

Answer: thank you.

Comment: However, it could be further strengthened by acknowledging the limitations of BCI as a measure of fitness, especially when discussing dispersal and the Chitty effect.

Answer: added after Line 429 ‘It is important to note that the body condition index is only a proxy for individual fitness. As a result, its value in explaining the Chitty effect may be limited.’

Comment: The introduction would be even stronger by explicitly stating the research question.

Answer: text expanded.

Comment: Consider mentioning the limitations of using BCI as a proxy for fitness. Briefly acknowledge that other factors can influence fitness besides body condition.

Answer: added to discussion ‘One can consider other factors than BCI that influence small mammal fitness, such as pollution, genetic factors, behavioral factors, health and diseases, etc. However, we have no information about these factors in our study.’

Comment: Briefly mention the study methods planned to address the research question.

Answer: added to Line 139 ‘we focused our analysis to inter- and intra-species aspects we focused our analysis to inter- and intra-species aspects’

Comment: L41: Replace "BMI" with "BCI" (Body Condition Index) as BMI is used for humans.

Answer: apologies for oversight, this was autocorrection.

Comment: L50: Consider rephrasing "The Dehnel phenomenon... explains..." to "Small shrews survive winter by..." for a smoother flow.

Answer: rephrased as suggested.

Comment: L52: "Voles increase chances of survival..." - Consider "by accumulating fat reserves..." to clarify the strategy.

Answer: no no, they do not accumulate fat – instead, they keep not too big body mass to lessen food requirements during winter time. It seems strange, but herbivores require this, so we added some text to explain.

Comment: L57: "Body condition may play a role..." - "May" could be strengthened to "likely plays" since the introduction has established the importance of BCI.

Answer: rephrased as suggested.

Comment: L68: "Various factors were analyzed..." - Consider "Several studies have analyzed..." for a more natural flow.

Answer: rephrased as suggested.

Comment: L75: "Schweiger et al. [27] caution..." - Consider "Some researchers caution..." for better flow.

Answer: rephrased as suggested.

Comment: L92: "Other references are..." - This sentence can be omitted because the focus has shifted to the study itself.

Answer: deleted as proposed.

Comment: L93: "yellow-necked mouse" - This is the first mention. Add the full scientific name.

Answer: there is Latin name in Line 93

Comment: Finally, the introduction becomes long and boring. Please keep it short and focus on your topic. In fact, the introduction is more like a discussion than an introduction.

Answer: we shortened text, however, we intended to show what is known about the species analyzed in the manuscript.

Method

Comment: Habitat is mentioned as a factor influencing BCI in the introduction, but then excluded from the analysis here (sentence 155). This is a significant decision and needs justification.

Answer: results of habitat analysis are already presented as separate publication. We cannot duplicate these, and, material is too big to be analyzed in one paper.

Comment: The authors acknowledge unequal trapping effort across habitats (sentence 147). This could bias the results towards habitats with more trapping effort.

Answer: we did rarefaction analysis to see, if sample sizes might be too small for analyses. Dominant species are well-presented in all habitats, while rare species were not trapped in many habitats.

Comment: The total number of individuals trapped in fragmented habitats is mentioned (1181), but the distribution across forest, meadow etc. within fragmented habitats is not provided.

Answer: this cannot be specified, as dissecting protocols were not related to single trap and it’s habitat.

Comment: The total number of individuals trapped (28,620) doesn't match the sum of individuals listed by specific habitat types (total = 26,442). There's a discrepancy of 2,178 individuals for unaccounted habitats.

Answer: we are extremely grateful for noticing this – discrepancy was due to exclusion of two rat species and unidentified voles, and we (my bad :) forgot to correct habitat data. Analyses are correct.

Comment: The exclusion of some species (brown rat, black rat, and unidentified voles) is mentioned (sentence 166) but not justified. Were there health concerns, identification issues, or other reasons?

Answer: rats were trapped under different protocols, sometimes obtained from farm owners, so we cannot be sure about their BCI (animals might be sitting in live traps for some time, etc.). explanation added.

Comment: While both balance and electronic scales are mentioned (sentence 192), it would be good to clarify if there were any systematic differences in measurements between the two types.

Answer: unfortunately, we did not check differences, but accuracy in all scale types was no bigger than 0.1 g. depending on this, we excluded scale types from text.

Comment: The results show contrasting patterns of BCI and age across species. While some species show increasing BCI with age, others show a decrease. The text acknowledges this variation but doesn't provide a strong explanation for the differences (sentences 278-279).

Answer: unfortunately, we have no stringer explanation, as this is the first such study done on retrospective data. Maybe later we will relate BCI with results of isotopic analyses of the age-related diet differences.

Comment: The results acknowledge that the sex-specific BCI difference observed in S. betulina might be unreliable due to a small sample size (sentence 298).

Answer: we tried to say so (not confirmed by post-hoc …), maybe change ‘but’ to ‘so’ will help?

Comment: The results show contrasting patterns of BCI and reproductive status across species. For example, breeding can increase or decrease BCI depending on the species (Figures 5, 6, 7). The text acknowledges this variation but could benefit from a more thorough discussion of the potential biological mechanisms (e.g., energetic demands of reproduction) underlying these differences.

Answer: unfortunately, we have no data how energetic demands of reproduction differs in small mammals, especially between their species. Maybe our paper will start discussion with the other researchers?

Comment: L146: "Material was collected..." - Consider "Small mammals were trapped..." for a clearer start.

Answer: replaced as suggested, thank you.

Comment: L148: "Due to several decades-old laboratory protocols..." - This justification for unequal trapping effort could be explained more concisely.

Answer: in the old protocols, some details could be missing, such as to a habitat of every trapped individual when one line of traps was set in a fragmented habitats. We tried to re-word for clarity.

Comment: 151: "fragmented habitats, including forests, meadows, wetlands, and agricul- 152 tural lands" - This could be rephrased: "including a mix of forest, meadow, wetland and agricultural land."

Answer: replaced as suggested.

Comment: L160: "The captured individuals..." - Consider "Small mammals..." for better flow.

Answer: replaced as suggested.

Comment: L192: "balance, spring" - Consider replacing with "balance scales, spring scales" for clarity.

Answer: we excluded scale type, as their accuracy was the same as 0.1 g of better, therefore did not influenced measurements.

Comment: L193: "standard measures... were taken...to the nearest 0.1 mm" - This could be rephrased for conciseness: "Standard body measurements (mass, length, tail length, hind foot length, ear length) were recorded to the nearest 0.1 mm using mechanical or electronic calipers."

Answer: replaced as suggested.

Comment: Consider "Over 80% of the measurements were conducted by the same person..."

Answer: replaced as suggested.

Comment: L203: "The age groups and reproductive status..." - Consider "Age groups and reproductive status..."

Answer: replaced as suggested.

Comment: L206: "Body mass as an indicator..." - This could be rephrased: "Body mass was only used as an indicator of age class when the two previous parameters were unavailable."

Answer: replaced as suggested.

Comment: L 210-216: Consider combining these sentences into a table for clarity

Answer: we did as advised.

Comment: L  223-229: This section describing how litters were identified could be condensed and rephrased for improved readability.

Answer: this is very important question, as references for this are not presenting the same information, therefore we were unable to shorten the text.

Comment: L  241: "Gaussian family GLM..." - Consider "Gaussian generalized linear model (GLM)..." for brevity.

Answer: maybe we do not understand you correctly, as proposed text is not shorter. Therefore we excluded all remaining text to the end of paragraph, and changed other text accordingly.

Results

Comment: The results acknowledge that M. avellanarius and A. amphibius (species with high BCI) are underrepresented in the sample (sentence 265). While they are excluded from further analysis, consider briefly discussing the potential consequences of this exclusion on the overall findings.

Answer: after your comment we excluded this sentence as not needed (data are still presented in Supplements, even if age- or sex- groups are missing for some species). Again, thank you for noticing this.

Comment: L253: "All tested factors..." - This could be rephrased for better flow: "All factors examined were significantly related..."

Answer: rephrased as advised

Comment: L261: "rather wide range" - Consider replacing with a more specific descriptive statistic, e.g., "standard deviation".

Answer: we mention namely range, that is, difference between minimum and maximum.

Comment: L266: "analyzed further" - Consider replacing with "included in further analysis".

Answer: sentence was excluded, see above comment

Comment: L269: Consider rephrasing "2.82< BCI < 3.0" to "BCI between 2.82 and 3.0".

Answer: rephrased as advised

Comment: L276: "We found significant differences..." - This could be rephrased for conciseness: "Significant differences in BCI..."

Answer: Yes, but then we would need to add ‘were observed’, as otherwise sentence is not clear,

Comment: L280: "all differences between age groups..." - Consider "BCI differed significantly between all age groups..."

Answer: rephrased as advised.

Comment: L283: Consider rephrasing "although the BCI was similar in the latter two age categories" to "but not between subadults and adults".

Answer: rephrased as advised.

Comment: Figure references: Double-check the numbering of supplementary figures (Figure S1 mentioned in sentence 286, but Figure 2 mentioned in sentence 292).

Answer: numbering and referring both are correct: fig. S1 is for age related differences, Fig. S2 – for sex-related differences.

Comment: L309: "Participation in the reproductive processes..." - This could be rephrased for conciseness: "Reproductive status..."

Answer: rephrased as advised.

Comment: L314: "as in numerous other species..." - Consider replacing with "as observed in many other species" for better flow.

Answer: rephrased as advised.

Comment: Ensure consistent numbering for supplementary figures (e.g., Figure S3 mentioned multiple times).

Answer: yes, but please note that there are parts of Figure S3, so these are mentioned where required.

Discussion

Comment: The discussion acknowledges contrasting patterns of BCI across species (e.g., granivores vs shrews, Figure 3) but could benefit from a more in-depth discussion of the underlying biological reasons.

Answer: with the limited references about BCI in small mammals, there are no more possibilities to expand discussion. In the future, if we will be able relate isotopic niche as a proxy of the diet, we might try to answer your comment better.

Comment: The discussion cites several studies to support findings, however, it might be beneficial to broaden the scope by incorporating a wider range of relevant literature on BCI and ecological factors.

Answer: number of references about BCI  in small mammals is really small, and this was a driving factor for our analyses.

Comment: L 352: "The average BCI..." - This could be rephrased for improved flow: "Small mammal BCI in Lithuania exhibits..."

Answer: rephrased as advised.

Comment: L 360: "Only one of these species, A. amphibius, is typical herbivore..." - Consider "Only A. amphibius among these species is a typical herbivore..."

Answer: rephrased as advised.

Comment: L 373: "When comparing the BCI..." - This sentence could be restructured for clarity: "When BCI is compared between age groups..."

Answer: rephrased as advised.

Comment: L 382: "such differences..." - Consider replacing with "similar differences" for better coherence with the previous sentence.

Answer: rephrased as advised.

Comment: L 400: "On the other hand..." - This could be replaced with a transition indicating a contrasting point, like "In contrast..."

Answer: rephrased as advised.

Conclusions

Comment: No limitations of the study are mentioned.

Answer: we added limitations of the study.

Comment: The conclusion could benefit from a stronger statement about the novelty of the findings, especially regarding the multi-species approach at mid-latitudes (mentioned in the introduction).

Answer: this is the very first sentence of conclusions.

Comment: Please remove the conclusion from the state of cutting the sentences and let the sentences continue.

Answer: there are now four paragraphs, we removed part of breaks.

Comments on the Quality of English Language

Comments are the same as above.

Answer: thank you very much for all your elaborated comments. We incorporated all your suggestions into the text, improving readability.

Round 2

Reviewer 2 Report

Comments and Suggestions for Authors

no comments.